# Holothurian Wall Hydrolysate Ameliorates Cyclophosphamide-Induced Immunocompromised Mice via Regulating Immune Response and Improving Gut Microbiota

**DOI:** 10.3390/ijms241612583

**Published:** 2023-08-09

**Authors:** Chen Yan, Huiru Qu, Xinli Li, Bin Feng

**Affiliations:** Department of Biotechnology, College of Basic Medical Sciences, Dalian Medical University, Dalian 116044, China; yanchen94r@gmail.com (C.Y.); quhuiru0613@163.com (H.Q.)

**Keywords:** holothurian wall hydrolysate, immunodeficiency, tight junction proteins, pro-inflammatory cytokines, microbial composition

## Abstract

Some biologically active compounds isolated from sea cucumbers stimulate the body’s immune response by activating immune cells. Immune function is closely related to the integrity intestinal barrier and balanced gut microbiota. However, it is unknown whether the daily administration of holothurian wall hydrolysate (HWH) ameliorated intestinal dysbiosis and barrier injury induced by immunodeficiency. This study aimed to investigate the immunomodulatory effect and the underlying mechanism of HWH in cyclophosphamide (CTX)-induced immunocompromised mice. BALB/c mice received CTX (80 mg/kg, intraperitoneally) once a day for 3 days to induce immunodeficiency, and then they received the oral administration of HWH (80 or 240 mg/kg) or levamisole hydrochloride (LH, 40 mg/kg, positive control), respectively, once a day for 7 days. We utilized 16S rRNA sequencing for microbial composition alterations, histopathological analysis for splenic and colonic morphology, Western blotting for expressions of tight junction proteins (TJs), and quantitative real-time (qRT)-PCR for measurements of pro-inflammatory cytokines. HWH attenuated the immune organ damage induced by CTX, increased the secretions of interleukin (IL)-6, IL-1β, and tumor necrosis factor (TNF)-α, and promoted the recovery of goblet cells and the production of TJs (claudin-1, occludin, and ZO-1) in the colon of the immunocompromised mice. Moreover, HWH promoted the growth of beneficial microorganisms such as *Lactobacillus*, *Lachnospiraceae*, *Christensenellaceae*, and *Bifidobacterium*, while it suppressed the populations of *Ruminococcus*, *Staphylococcus*, and *Streptococcus*. These results demonstrate that HWH elicits intestinal mucosal immunity, repairs the damage to intestinal mucosal integrity, and normalizes the imbalanced intestinal microbial profiles in immunocompromised mice. It may be helpful to identify the biological activities of HWH to support its potential use in new prebiotics, immunomodulatory agents, and medical additives for intestinal repair.

## 1. Introduction

The human gastrointestinal (GI) tract is colonized by a huge number of microorganisms, termed gut microbiota [1]. Healthy gut microbiota contain *Bacteroidetes* and *Firmicutes* [2], followed by *Actinobacteria*, *Proteobacteria*, and *Verrucomicrobia* [3], as well as methanogenic archaea, eukaryotes, and various phages [4]. Multiple host-endogenous and host-exogenous factors exert effects on gut microbiota to create a resilient, stable, and balanced microflora [5]. Individuals contain unique intestine flora that distinguish them from the population. They can be identified uniquely among populations of 100s based on microbiomes using metagenomic codes [6].

Once there are alterations to the influencing factors, such as genetic mutations, dietary preference, and antibiotics administration, the dysbiosis of gut microbiota occurs. Dysbiosis can induce the dysfunction of the intestinal barrier and the influx of pro-inflammatory bacterial fragments and low-grade chronic systemic inflammation [7]. Various studies have shown associations between dysbacteriosis and obesity [8], type 2 diabetes [9], hypertension [10], and inflammatory bowel disease [11].

Gut microbiota can prevent exogenous infections through antimicrobial agent production, nutrient competition, intestinal barrier integrity improvement, bacteriophage utilization, and immune response activation [12]. Additionally, gut microbiota also occupy binding sites in the mucus layer, covering enterocytes, and they prevent pathogens from colonizing the intestine of humans [13]. The intestinal epithelial barrier and immune system contribute to gut barrier integrity, thereby preventing risk factors and maintaining a symbiotic relation with other bacteria [14,15]. The intestinal epithelial barrier contains several types of intestine epithelial cells (IEC). IECs form a strong barrier using adherens junctions, tight junctions, and desmosomes, which prevent the leakage of intracellular components and the influx of exogenous pathogens through the intracellular space. The lamina propria of the epithelial layer promotes healthy communication between gut microbiota and immune cells [15]. Moreover, intestinal immune cells (e.g., dendritic cells, T cells, B cells, and macrophages) are related to IECs closely, maintaining intestinal homeostasis [16,17].

There is a crucial relationship between the compromised immune function and the integrity intestinal barrier. Compromised immune function leads to an impaired intestinal mucus barrier and disturbed gut microbiota [18,19]. Recently, immunomodulators for impaired immune function were developed mainly to stimulate impaired immunity, not to repair intestinal mucosal damage and not to improve the health of gut microbiota. For example, (E)-phenethyl 3-(3,5-dihydroxy-4-isopropylphenyl) acrylate gels exerted immunotherapeutic effects by modulating the balance of Th1/Th2/Th17/Treg cell subsets in allergic contact hypersensitivity [20]. Tiepishihu Xiyangshen granules showed prominent immunomodulatory activities by regulating TLR4/MAPK and PI3K/AKT/FOXO3a signal pathways [21]. However, gut microbiota and the intestinal barrier have important roles in chronic human diseases, which should not be ignored [22]. Therefore, the research focusing on effective methods to improve intestinal barrier function and repair the gut microbiota dysbiosis for immunodeficiency treatment is of great importance.

There is a long history of treating and preventing disease with natural products. Natural products, especially from terrestrial creatures, have long been a traditional source of therapeutic components [23]. Many researchers consider marine invertebrates more fruitful sources of novel therapeutic agents than any group of terrestrial species, and marine compounds have gradually attracted attention in recent years because of their unique structural and biological activities with less side effects [24,25]. Sea cucumbers (class Holothuroidea), belonging to the phylum Echinodermata, are precious seafood and important sources of medicine. The main active ingredients are polysaccharides, peptides, triterpene glycosides, etc., which have several important functions in the body, such as ameliorating glucose intolerance, inhibiting the growth of tumors, regulating the gut microbiota, and stimulating immunity [26,27,28,29,30,31]. Moreover, the latest study showed that *Holothuria leucospilota* polysaccharides played a role in treating constipation by optimizing intestinal flora composition, promoting intestinal peristalsis, relieving intestinal inflammation, and regulating intestinal electrolyte metabolism [32]. As a marine natural compound, sodium oligomannate (GV-971) therapeutically remodeled gut microbiota and suppressed gut bacterial amino acid-shaped neuroinflammation to inhibit Alzheimer’s disease [33]. Our previous work [34] also indicated that chitosan fought against ulcerative colitis in mice by mitigating intestinal microflora dysbiosis and regulating the expressions of TNF-α, claudin-1, occludin, and ZO-1. There is a potential mechanistic link between gut microbiota dysbiosis, intestinal barrier function, and diseases.

In recent years, the active substances from sea cucumbers were shown to be effective inductors of cellular immunity. However, there are few reports on the role of hydrolysates from sea cucumbers in intestinal immunity; the underlying mechanism between holothurian wall hydrolysate (HWH) and the immune system remains unclear. Therefore, in the present study, a CTX-induced immunocompromised mouse model was established to evaluate the immunomodulatory effect and the underlying mechanism of HWH treatment. The bacterial composition was altered after CTX administration and HWH treatment by 16S rRNA sequencing. Furthermore, histopathological analysis was performed followed by TJ (claudin-1, occludin and ZO-1) and pro-inflammatory cytokine (TNF-α, IL-6, and IL-1β) analysis. Our focuses were to investigate the impact of CTX on the intestinal barrier integrity and gut microbiota; to probe the potential of HWH as an immunomodulator in the amelioration of intestinal dysbiosis and the restoration of intestinal barrier impairment in immunodeficiency; and to provide novel insight for enhancing the value of the sea cucumber.

## 2. Results

### 2.1. Peptide Sequence Analysis of HWH

The total protein content, carbohydrate content, moisture content, and ash content of HWH were 91.93 ± 0.42%, 0.76 ± 0.05%, 2.60 ± 0.05%, and 2.03 ± 0.06%, respectively. LC-MS was used to obtain the total ion flow diagram of HWH (Appendix A). The peptide sequence identification result of HWH is shown in Table 1. The peptides detected in HWH were input to PeptideRanker for activity evaluation. A total of 23 peptides were identified in HWH; thereinto, the activity scores of five peptides (SRGLLSCLF, GFDGPEGPRGPPGSE, RGPAGPTGPTGPA, AAVAAAVAPPSPPPIAGPP, and FDGPEGPRGPPGSE) were >0.5. The peptides with high activity in HWH in molecular weight were 1051.55, 1454.64, 1134.57, 1649.91, and 1397.62 Da, respectively, and were mainly composed of peptides with 9–19 amino acids. These results showed that peptides were the main components of HWH.

### 2.2. Effects of HWH on Spleen Index and Spleen Histopathology

Histopathological images of a spleen are shown in Figure 1A. In the NC and N240 groups, the red pulp and white pulp were clearly distinguished. The section of spleen in the CTX group showed an unclear border between red pulp and white pulp and abnormal tissue morphology. However, the boundary region between white pulp and red pulp in the spleen was more apparent in the HWH80 and HWH240 groups. Additionally, the HWH240 group demonstrated smaller spleen damage than that of the HWH80 group due to a clearer boundary between red pulp and white pulp, which was closer to the NC group. In addition, compared with the CTX group, the reduced damage in the HWH80 and HWH240 groups reflected that HWH could alleviate CTX-induced injury in the spleen.

As shown in Figure 1B, the spleen indices of the CTX group were significantly decreased compared to those of the NC group. Compared to the CTX group, the HWH80, HWH240, and PC groups showed a drastic increase in the spleen indices (*p* < 0.001), whereas there was no significant difference in the spleen indices of mice between the NC and N240 groups. HWH treatment effectively increased the spleen indices (*p* < 0.05), which implied that HWH could reverse the CTX-induced immune dysfunction of the spleen.

### 2.3. Effects of HWH on Histopathology in Colon

The histological analysis of colonic tissues was performed by H&E staining (Figure 2A). The NC group demonstrated a normal histology consisting of a well-shaped and compact crypt containing goblet cells. In contrast, CTX-treated mice showed histological alterations consisting of a blunt and short crypt with an obvious infiltration of inflammatory cells. However, HWH80, HWH240, and PC groups exhibited the reduced infiltration of inflammatory cells and an improved colonic tissue structure, which was more similar to the NC group. The N240 group also showed a normal and regular colonic wall structure lined with dark-stained nuclei in the lamina propria.

PAS-positive goblet cells were observed in the crypts of the colon (Figure 2A). The OD of the CTX group was significantly lower than that of the NC group (*p* < 0.05), whereas the OD of the HWH80, HWH240, and PC groups was higher than that of CTX group significantly (*p* < 0.05) (Figure 2B). In addition, there was no significant difference between the NC and N240 groups. The results indicated that HWH could repair CTX-induced damage in the colon.

### 2.4. Effects of HWH on Intestinal Tight Junction Proteins

Compared to the NC group, reduced expression of TJs was observed in the CTX group (Figure 3). HWH intervention enhanced the expression of TJs significantly in the HWH80 and HWH240 groups compared with the CTX group (*p* < 0.05), whereas there was no significant difference between the NC and N240 group. These findings suggested that HWH could improve the expressions of TJs and reduce the damage in the colon caused by CTX-induced immunodeficiency. 

### 2.5. Effects of HWH on Pro-Inflammatory Cytokines Levels

Compared to the NC group, CTX reduced the levels of inflammatory cytokines (Figure 4). The HWH80, HWH240, and PC groups showed a significant increase compared to the CTX group in the levels of IL-6 and IL-1β (*p* < 0.05), whereas merely the HWH240 group showed a significant increase compared to the CTX group in the level of TNF-α. Compared to the NC group, the significant increase of the N240 group indicated that HWH could improve the secretion of TNF-α. CTX induced a significant decrease of secretion of IL-6, implying the inhibitory effects of CTX on IL-6. These findings reflect that HWH reversed the CTX-induced decreased secretion of pro-inflammatory cytokines.

In addition, the HWH80, HWH240, and PC groups showed higher secretions of IL-1β compared to the NC group significantly, while HWH240 demonstrated a significant increase in the secretion of IL-6, which indicated that HWH could increase the secretions of IL-1β and IL-6 in normal mice. The results reflected that HWH also promoted the secretion of immune cytokines.

### 2.6. Effects of HWH on the Overall Composition of Gut Microbiota

The gene sequencing of 16S rRNA was used to detect the gut microbial composition. Illumina pair-end sequencing returned 3,790,530 raw sequences across 18 fecal samples to investigate the dynamic alterations of the microbial composition in the gut. After paired-end read assemblies and quality filtering, a total of 1,859,229 sequences were used in the downstream analysis. The Rank abundance curve shown in Figure 5A suggests that the current sequencing depth already found most of the microbial phylotypes. The richness and diversity in all the six groups were also observed by rank abundance, which showed high species richness and evenness in HWH-treated groups in general. Clustering at 97% identity produced 147,549 unique operational taxonomic units (OTUs) in total and 81,697 OTUs on average for each sample (ranging from 57,439 to 105,717), providing over 99.9% coverage of all samples. The details of sequences of all samples are shown in Appendix A. A box plot was plotted to indicate higher α-diversity in the HWH-H and N240 groups (Figure 5B). The results of Chao1, ACE, and OTU in HWH-treated groups were higher than that in the CTX group (M) (Appendix A), indicating the diversity of species that was decreased due to CTX. These results conclude that CTX obviously reduced the overall microbial population in the gut. HWH treatment helped to partially restore the community of gut microbiota.

### 2.7. Effects of HWH on Beta Diversity of Gut Microbiota

The beta diversity was determined by principal component analysis (PCA) and principal coordinate analysis (PCoA), which revealed bacterial community structural variation in all groups (Figure 6). The top two principal components in PCA analysis occupied 14.95% and 52.52% of the variation of the overall data. This showed that the structure of the microbial community in the CTX group (M) was significantly different from that of the NC group. PCoA analysis demonstrated that the microbial community structure in the CTX group (M) was clustered distantly from the NC group, and the HWH-H group showed a much higher difference from the CTX group than that of the HWH-L group. The PC group was close to the HWH-H group. These data were consistent with the α-diversity.

### 2.8. Effects of HWH on Taxonomical Analysis of Gut Microbiota

At the phylum level, the OTUs were classified as 11 phyla. The gut consists of *Bacteroidetes* and *Firmicutes*, followed by *Proteobacteria*, *Actinobacteria*, *Deferribacteres*, *Chloroflexi*, *Cyanobacteria*, *Gemmatimonadetes*, *Saccharibacteria*, *Tenericutes*, and *Verrucomicrobia*. Compared with the NC group, there was a decrease in *Firmicutes* (M: 74.18% vs. NC: 28.34%) and an increase in *Bacteroidetes* (M: 69.25% vs. NC: 18.3%). In addition, the HWH-L group demonstrated an increase in the abundance of Actinobacteria and drops in the proportions of *Proteobacteria* and *Saccharibacteria* compared to the CTX group (M) (Appendix A). CTX treatment decreased the proportion of *Firmicutes* compared to the NC group, whereas the HWH intervention increased the abundance of *Firmicutes* compared to M group (Figure 7A). At genus level, CTX treatment increased the abundance of *Ruminococcus* and decreased *Lactobacillus*, *Lachnospiraceae*, and *Bifidobacterium*. In contrast, HWH intervention reversed these changes, *Ruminococcus* downregulated, and *Lactobacillus*, *Lachnospiraceae*, and *Bifidobacterium* upregulated (Figure 7B). These outcomes imply that HWH treatment improved the gut microbiota community in CTX-induced immunocompromised mice.

### 2.9. Effects of HWH on Phylotypes of Gut Microbiota

LDA results (Figure 8A) demonstrated twenty discriminative features in the NC group, and *Bacillus* and *Lactobacillus* were the main microbiota. The CTX group (M) showed ten dominant microorganisms as well, and the major microbes were *Clostridiales*, *Staphylococcus*, and *Streptococcus*. The HWH-L group displayed two key microorganisms, *Prevotellaceae* and *Christensenellaceae.* The HWH-H group displayed eleven discriminative microbes, including *Clostridia*, *Clostridiales*, *Lactobacillus*, etc. The PC group showed five dominant microorganisms, and *Rikenellaceae* and *Alistipes* were the major microbiota. The N240 group (NH) exhibited five dominant phylotypes, including *Bacteroidales*. In the cladogram (Figure 8B), *Lactobacillus* had the highest abundance in the purple part, which represented the NC group. *Clostridiales* exhibited the highest abundance in the marine blue part, which represented the HWH-H group. Overall, these findings suggest that HWH intervention altered the key phylotypes of gut microbiota in CTX-treated mice and promoted the growth of specific bacteria.

## 3. Discussion

Metabolites of CTX can produce cytotoxic activity. CTX alters the microbial community composition in the colon and damages intestinal mucosal immunity [35]. It was reported that immunosuppression could damage the intestinal mucosal integrity by decreasing the expression of TJs [36]. Those previous findings indicated that a unipotent immune stimulator could not match immunocompromised hosts, and it was required to find a multipotent immune agonist with properties normalizing the gut dysbiosis and repairing the impaired intestinal mucosa. 

Some biologically active compounds isolated from sea cucumbers stimulated the body’s immune response by activating immune cells. An acidic mucopolysaccharide effectively inhibited the growth of hepatocellular carcinoma through the stimulation of immune organs and tissue proliferation, leading to the enhancement of cellular immunity pathways [37]. Additionally, sea cucumber peptides exhibited potential antiallergic activities by upregulating the immune response of T lymphocyte subpopulations [38]. In the studies of the immune system, spleen indices are significant indicators that reflect immune functions because the spleen is an important immune organ [39]. In addition to its ability to promote splenic lymphocyte proliferation, spleen indices serve as a parameter to indicate splenic immune function. *Apostichopus japonicus* glycosaminoglycan promoted lymphocyte proliferation in the spleen [40]. The spleen index of ovalbumin-induced allergic mice was significantly decreased compared to that of normal mice, but this index was significantly increased with sea cucumber peptide treatment [38]. Our findings showed that the immune dysfunction induced by CTX was reflected by a significant drop in the spleen indices, which were reversed by HWH intervention.

Cytokines are secreted by immunocytes and non-immune cells that regulate immune functions. It has been reported that CTX-induced immunosuppression could inhibit the secretions of immune cytokines, including TNF-α [41], IL-6 [42], and IL-1β [43]. TNF-α killed tumor cells directly and promoted the formation of cytotoxic lymphocytes and the activity of macrophages [36]. IL-6, a pyrogen similar to IL-1 and TNF-α, mediated the acute phase response characterized by leukocytosis and increased acute phase reactants [44]. IL-1β, the major endogenous pyrogen, possessed multiple properties in response to infection, injury, and immune alteration [45]. The production of cytokines, specifically TNF-α, IL-6, and IL-1β in macrophage cell line RAW264.7, was enhanced by water-soluble sulfated fucan from the sea cucumber A. leucoprocta [46]. Likewise, the expressions of TNF-α and IL-6 were upregulated by *Holothuria leucospilota* polysaccharides in CTX-induced immunosuppressed mice [47]. Secretions of TNF-α, IL-6, and IL-1β in the colon were investigated in this study. The results demonstrated that CTX-treated mice showed a significant decline in the generation of IL-6. The administration of HWH (240 mg/kg) showed increased secretions of TNF-α, IL-6, and IL-1β compared with the CTX group, whereas HWH 80 and PC groups exhibited significantly increased secretions of IL-6 and IL-1β. These results are consistent with those in the previous studies. Overall, HWH could stimulate immune function by increasing the secretions of cytokines.

Mucus, which is secreted by goblet cells in the colon, serves as a crucial protective indicator against pathogens [48]. Goblet cells can synthesize and secrete mucin to the intestinal lumen. Furthermore, mucins are a family of large, complex, glycosylated proteins, and the glycan chain of the mucin could be stained by PAS [49,50]. After fermentation, *Holothuria leucospilota* polysaccharides were observed to increase the goblet cell count in histological analysis [51]. To investigate the effect of HWH intervention on the CTX-induced immunosuppression driven intestinal mucosal injury, we elucidated the changes in colon histology, intestinal permeability, and microbial composition. The results demonstrated that CTX-induced immunosuppression altered the gut morphology. The histological analysis exhibited inflammatory cell infiltration, loss of crypts, and lower goblet cells in the CTX-treated group, whereas HWH treatment improved gut morphology. In addition, the PAS-staining section showed that decreased mucin in the goblet cells was observed in the CTX group, while HWH intervention promoted the expression of mucins in the goblet cells.

TJs in intestinal epithelial cells consisted of different components, including claudin, occludin, and ZO-1 [52], which maintained the integrity of the intestinal barrier [53,54]. A previous study observed that the dietary fucoidan of *Acaudina molpadioides* could increase the mRNA of occludin in CTX-treated mice intestines [55]. Although there is limited research about the effects of sea cucumber on TJs, the correlation between intestinal immune function and intestinal integrity is widely acknowledged [56]. Therefore, we investigated the effects of HWH on TJs in this study. Results showed that CTX-induced immunosuppression led to decreased expressions of TJs, which were reversed by the administration of HWH, indicating that HWH can upregulate the expression of occludin. In the assessments of claudin and ZO-1, the CTX group displayed a significant drop in the expressions compared to the NC group. HWH80, HWH240, and PC groups upregulated the expressions of claudin and ZO-1 significantly compared to the CTX group, suggesting that HWH can repair the immunosuppression-induced damage to intestinal physical integrity. 

Gut microbiota played crucial roles in modulating the host’s immune system [57]. Recently, a variety of natural products have been proven to have the ability of regulating gut microbiota [58]. Sulfated polysaccharides benefit the host health via modulating gut microbiota composition [31]. A sea cucumber polysaccharide reduced the intestinal barrier damage, inhibited the inflammatory response, and improved the intestinal microbiota, which was relevant to the production of SCFA [32]. Long-chain bases from sea cucumber could significantly increase the Shannon index and decrease the Simpson index, implying the increased diversity of the gut microbiota [59]. Meanwhile, sulfated polysaccharide could also improve the Chao1 and Ace indexes, implying improved richness and diversity of gut microbiota [60]. We investigated the effect of HWH on the CTX-treated mice gut microbiota with 16S rRNA sequencing. HWH ameliorated gut dysbiosis in CTX-induced immunosuppression via alpha diversity analysis. Chao and ACE indices increased in the HWH group compared to the CTX croup. Moreover, the number of OTUs also exhibited an increasing trend of microbial community diversity after HWH administration compared to the CTX group. Beta diversity analysis (PCA and PCoA) revealed that there was an obvious difference between NC and CTX groups in the overall microbial community structure. Similar observations could be seen between the CTX and HWH-H groups.

*Firmicutes* and *Bacteroidetes* dominated the gut microbiotas [61], accounting for more than 90% of the microbiota. In the NC group, the proportion of *Firmicutes* was greater than that of *Bacteroidetes*, whereas the CTX disturbed the gut microflora, obviously, and lowered the proportion of *Firmicutes*. However, HWH intervention increased the proportion of *Firmicutes* compared to the CTX group, resulting in an increased *Firmicutes*/*Bacteroidetes* (F/B) ratio. A recent study has reported that high salt feeding mice contained a reduced F/B and a disrupted immunological response characterized by disturbed immune-related gene expression in the colon [62], which might account for the decreased F/B in the CTX-induced immunosuppression. In addition, the HWH-L group demonstrated a rise in the abundance of *Actinobacteria* and a decline in the proportions of *Proteobacteria*, a microbial marker of gut dysbiosis [63], and *Saccharibacteria* compared with the CTX group. The administration of fucoidan extracted from the sea cucumber *Pearsonothuria graeffei* could lead to an increase in the abundance of *Bacteroidetes* and *Actinobacteria*, while decreasing the richness of *Firmicutes* and *Proteobacteria* [64]. *Saccharibacteria*, known as TM7, has been reported to have an association with inflammatory bowel diseases (IBDs). It plays an important role as a promoter of inflammation in the early stages of inflammatory mucosal processes by triggering inflammation directly or modifying growth conditions for competing bacterial populations [65]. *Bifidobacterium* and *Lactobacillus* were widely used as benchmarks for evaluating the effect of prebiotics, containing lots of beneficial properties for the host. For example, both RG I (potato galactan-rich rhamnogalacturonan I) and its corresponding oligosaccharides/oligomers showed their prebiotic properties by stimulating the growth of *Bifidobacterium* spp. and *Lactobacillus* spp. [66]. *Lactobacillus* was promoted with chitosan treatment in DSS-induced ulcerative colitis mice [34], while *Ruminococcus* are usually linked to IBDs. Previous reports revealed how some strains of *Ruminococcus* could drive the inflammatory responses that characterize IBD [67]. The amount of *Enterobacteriaceae* and *Ruminococcus gnavus* increased, and *Faecalibacterium* and *Roseburia* disappeared in IBDs [68]. Similarly, in the rats’ obesity model, the relative abundance of *Romboutsia*, *Ruminococcus*, *Corynebacteriume*, and *Saccharibacteria* groups increased, brought about by a high-fat diet (HFD), and the above upregulation was significantly reversed by the intervention of winter melon and lotus leaf Tibetan tea [69]. Alterations in the abundance of microorganisms may cause changes in some metabolic pathways, in addition to altering the biosynthesis and degradation of some secondary metabolites, which are mostly associated with diseases. In our work, *Bifidobacterium* and *Lactobacillus* decreased after CTX treatment. HWH intervention improved the structure of the gut microbiota induced by CTX and increased *Bifidobacterium* and *Lactobacillus* compared to the CTX group, while significantly decreasing *Ruminococcus* and *Saccharibacteria*.

Moreover, LEfSe analysis results indicated that the key phylotypes in the NC group were *Lactobacillus*, *Lactobacillaceae*, and *Lactobacillales*, which belongs to beneficial bacteria of gut microbiota. *Lactobacillus* was particularly characterized in the protection from pathogenic bacteria, the modulation of the immune system to potentially reduce the risk of allergies and cancer, the reduction of radical oxidative species and cholesterol levels, and the potential benefit in diabetes [70]. *Lactobacillus* also strengthened intestinal barrier function, promoted TJ integrity, and protected against experimental necrotizing enterocolitis [71]. The discriminative microbes of the CTX group involved *Staphylococcus* and *Streptococcus*, which can serve as conditioned pathogens [72,73]. The above microbes may pose a serious threat for the human or animal host when they obtain access to inner layers of the body through breaches in the skin or membranes [74,75]. The family *Christensenellaceae* showed compelling associations with host health, which suggested that the cultured *Christensenellaceae* could be considered a therapeutic probiotic to improve human health [76]. The results demonstrated that the gut microbiota of the NC group were dominated by beneficial bacteria, such as *Lactobaccilus*, which disappeared in the CTX group. The harmful bacteria in the CTX group included *Staphylococcus* and *Streptococcus*, which vanished in the HWH-L and HWH-H groups. Then, *Christensenellaceae* dominated the gut microflora of the HWH-L group. Therefore, CTX caused a change in the gut microbiota composition, inhibited the growth of beneficial bacteria, and promoted the growth of harmful bacteria, while the intervention of HWH could reverse this trend. HWH had an excellent ability in regulating the disturbance of gut microbiota caused by CTX.

## 4. Materials and Methods

### 4.1. Materials and Reagents

HWH was supplied and analyzed by Zhen Jiu Co., Ltd. (Dalian, China). Cyclophosphamide was purchased from Aladdin (Shanghai, China). Levamisole hydrochloride was purchased from Shanxi Hongbao Veterinary Pharmaceutical Co., Ltd. (Yuncheng, China). The First Strand cDNA Synthesis Kit and the SYBR PrimeScript^™^ RT-PCR Kit were purchased from Takara Bio Co., Ltd. (Shiga, Japan). The primer sequences of TNF-α, IL-6, and IL-1β were synthesized by Sangon Biotech Co., Ltd. (Shanghai, China). The primary antibody against occludin was purchased from Proteintech Group Co., Ltd. (Wuhan, China). Antibodies against claudin and ZO-1 were from Wanleibio Co., Ltd. (Shenyang, China). E.Z.N.A.^®^ Stool DNA Kit was purchased from Omega Bio-Tek Co., Ltd. (Norcross, GA, USA). AxyPrep DNA Gel Extraction Kit was purchased from Axygen, Inc. (New York, NY, USA). 

### 4.2. Preparation of HWH

HWH was derived from sea cucumbers via enzymatic hydrolysis and produced by Dalian Zhenjiu Biological Industry Co., Ltd (Dalian, China). Briefly, the sea cucumbers were cleaned after gutting, thoroughly crushed by a blender, hydrolyzed with a neutral protease (3000 U/g), precipitated, filtrated, and dried to obtain HWH. 

### 4.3. Composition Determination of HWH

The moisture content of HWH was measured by using the direct drying method. The ash content of HWH was measured by using the high-temperature ashing method. The total protein content of HWH was determined by the Kjeldahl method [77]. The total carbohydrate content of HWH was determined by the phenol-H_2_SO_4_ method [78]. 

### 4.4. Identification of Peptides in HWH by LC-MS/MS

A 2% HWH solution was prepared and centrifuged in an ultrafiltration tube with an interception capacity of 10 kDa. Then, filtrate was collected and analyzed by the Easy-nLC1200-Orbitrap-MS/MS system (Thermo Fisher Scientific, Waltham, MA, USA). Chromatographic separation was performed on a reversed-phase C18 column (0.15 mm × 120 mm, 1.9 μm) at nano-flow gradient (flow rate, 600 nL min^−1^). The mobile phase A was composed of ultrapure water (containing 0.1% formic acid), and the mobile phase B was composed of acetonitrile (containing 0.1% formic acid) with a gradient program of 0–11 min, 93–85% A; 11–48 min, 85–75% A; 48–62 min, 75–60% A. The acquisition parameters consisted of one MS full scan (300–1400 *m*/*z*) in the Orbitrap mass analyzer with an automatic gain control (AGC) target of 5 × 10^5^, maximum injection time of 50 ms, and a resolution of 120,000 followed by a ddMS2 scan with an AGC target of 5000, maximum injection time of 35 ms, and one microscan. The intensity threshold to trigger an MS/MS scan was set to 1 × 10^20^ (maximum intensity) and 5000 (minimum intensity). Fragmentation occurred in the higher energy collision-induced dissociation (HCD) trap with collision energy (CE) set to 30% and stepped CE set to 5%. The dynamic exclusion was applied using a setting of 18 s. The ionic charge was set to 2–6. Maxquant software (v.2.0.1.0) was used for peptide matching. The measured peptides were evaluated by PeptideRanker.

### 4.5. Animals and Experimental Design

Female BALB/c mice (weight, 20 ± 2 g) were purchased from the Animal Experimental Center of Dalian Medical University, Dalian, China (Certificate of quality number: SCXK (Liao) 2018-0003; 20 May 2018). All experimental procedures were approved by the Animal Care and Research Ethics Committee of Dalian Medical University (Approval number: SYXK (Liao) 2018-0006; 18 November 2018). Mice were accommodated in an environmentally controlled room and had free access to standard diet and water ad libitum. After acclimation for one week, sixty mice were randomly divided into six groups (10 mice in each group), named NC, CTX, HWH80, HWH240, PC (positive control), and N240 (HWH control). Groups NC and N240 received normal saline and 240 mg/kg of HWH, respectively, and the other four groups received CTX (80 mg/kg, intraperitoneally) once a day for 3 days to induce immunodeficiency [30]. Then, groups CTX, HWH80, HWH240, and PC were treated with normal saline, 80 mg/kg, 240 mg/kg of HWH [26], and 40 mg/kg of levamisole hydrochloride [35] intragastrically once a day for 7 days. All animals were killed 12 h later to collect fecal samples for microbial composition analysis using 16S rRNA Illumina sequencing. Part of the colon and spleen tissues was fixed in 4% paraformaldehyde for histological analysis, and the remaining parts were prepared for Western blotting and qRT-PCR.

### 4.6. Determination of Spleen Index

The spleen index was calculated as follows:Spleen index = spleen weight (mg)/body weight (g) × 100%

### 4.7. Histological Analysis

The distal colon and spleen were dissected and fixed in 4% paraformaldehyde, respectively, for 24 h, gradient dehydrated by ethanol, vitrificated by xylene, and embedded in paraffin. Sections of 3 µm thickness were prepared via a microtome (Leica, Wetzlar, Germany). Spleen slides were stained with hematoxylin and eosin (H&E) after rehydration. Colon slides were stained with H&E and periodic acid–Schiff (PAS). The steps for H&E staining were as follows: (1) We performed three xylene treatments on slides in sequence, 15 min each. (2) We treated the prepared slides with concentration gradient ethanol and immersed them in absolute ethanol I, absolute ethanol II, and 95%, 85%, and 75% ethanol in turn for 5 min each, and we washed the slides with distilled water for 5 min. (3) We immersed the slides in hematoxylin staining solution for 5 min, then rinsed with tap water for 5 min, placed them in hydrochloric acid alcohol for a few seconds, rinsed with tap water for 5 min to turn blue, and then immersed the sections in eosin staining solution for 5 min. (4) We dehydrated them in ascending alcohol solutions (50%, 75%, 85%, 95% × 2, 100% × 2). (5) We cleared them with xylene I and II for 3 min each. (6) We mounted a coverslip onto the section of glass slide with neutral gum. The steps for PAS staining were as follows: (1) deparaffinized and hydrated to water; (2) oxidized in 0.5% periodic acid solution for 5 min; (3) rinsed in distilled water; (4) Placed in Schiff reagent for 15 min (sections became light pink color during this step); (5) washed in tap water for 5 min (immediately sections turned a dark pink color); (6) counterstained in Mayer’s hematoxylin for 1 min; (7) washed in tap water for 5 min; (8) dehydrated and coverslip using a synthetic mounting medium. The concentrations of hematoxylin and eosin were 0.5 and 1% (*w*/*v*), respectively. As for the volume of dye, slides were immersed in the staining box filled with about 50 mL of dye. The staining time for H&E was 5 min, while the time for the Schiff reagent was 15 min. Images were obtained using an optical microscope. Spleen and colon samples were observed at 40× and 100× magnification, respectively.

### 4.8. Western Blotting Analysis

Total protein was extracted from the colon and homogenized in RIPA buffer with 1:100 phenylmethylsulfonyl fluoride (PMSF) using ultrasonic extraction: work for 4 s and rest for 4 s (duration at 8 min and power of 10 watts), followed by resting on ice for 30 min, and centrifugation at 12,000 rpm for 20 min at 4 °C to obtain the supernatant. Protein concentration was determined using a BCA kit according to the manufacturer’s instructions. Total protein (20 µg) was fractionated on a 12% SDS-PAGE and transferred to PVDF membranes for 40 min for β-actin, 30 min for claudin-1, 60 min for occludin, and 180 min for ZO-1, respectively. PVDF membranes were blocked with 5% skimmed milk and incubated with primary antibodies at a 1:2000 dilution overnight at 4 °C, followed by incubation with HRP-conjugated secondary antibodies at a 1:8000 dilution for 2 h at room temperature after washed with TBST three times. Blots were visualized using the ECL kit (Thermo Fisher, Waltham, MA, USA), and images were captured by the ChemiDoc MP imaging system (Bio-Rad, Hercules, CA, USA). β-actin was used as an internal reference. 

### 4.9. Quantitative Real-Time PCR

Total RNA was extracted from colon tissue with TRIzol reagent, and cDNA was obtained by the First Strand cDNA Synthesis Kit according to the manufacturer’s protocol. Then, the qRT-PCR was performed using the SYBR PrimeScript^™^ RT-PCR Kit (Takara, Dalian, China). Data were analyzed using 2^−ΔΔCt^ method. GAPDH was used as an internal control. Primer sequences used are shown in Appendix A.

### 4.10. 16S rRNA Sequencing of Gut Microbiota

Total DNA was extracted from fecal samples using the E.Z.N.A.^®^ Stool Kit. The V3-V4 regions of bacterial 16S rRNA gene were amplified by PCR with primers 338F (5′-ACTCCTACGGGAGGCAGCA-3′) and 806R (5′-GGACTACHVGGGTWTCTAAT-3′) and the following protocol: initial denaturation at 98 °C for 1 min, followed by 30 cycles of denaturation at 98 °C for 10 s, annealing at 50 °C for 30 s, elongation at 72 °C for 1 min, and a final extension at 72 °C for 5 min. The PCR products were separated and detected on a 2% agarose gel, purified by AxyPrep DNA Gel Extraction Kit, and quantified with the Quantifluor dsDNA quantification system. The amplicons were normalized, pooled, and sequenced on the Illumina NovaSeq PE250 (Shanghai Origingene Biopharm Technology Co.,Ltd, Shanghai, China). The sequencing library was generated using the NEB Next^®^Ultra^™^DNA Library Prep Kit for Illumina (NEB, Ipswich, MA, USA). The library quality was assessed on the Qubit@2.0 Fluorometer (Thermo Scientific, Waltham, MA, USA) and Agilent Bioanalyzer 2100 system (Agilent, Santa Clara, CA, USA). Quality sequences were acquired by removing low-quality reads. The taxonomic assignment was performed using the SILVA 132. Operational taxonomic units (OTUs) were generated using Usearch version 10 with a dissimilarity cutoff of 0.03. Alpha diversity was calculated with mothur v.1.30, and beta diversity was analyzed using Quantitative Insights into Microbial Ecology (QIIME) with weighted and unweighted Unifrac distance matrix to measure similarity in microbial composition between samples. Bacterial relative abundance was analyzed and performed by QIIME (v.1.8.0) and R software (v.3.2.5). The linear discriminant analysis (LDA) and LDA effect size (LEfSe) methods were used to quantify biomarkers within different groups with statistical significance.

### 4.11. Statistical Analysis

GraphPad Prism 8.0.1 (La Jolla, CA, USA) was used for analysis. A one-way analysis of variance (ANOVA) was performed with Tukey’s multiple comparison test to determine the significance of differences, and a *p*-value < 0.05 was considered significant.

## 5. Conclusions

HWH can improve the immune function, repair the impaired intestinal mucosal integrity, and regulate the diversity and composition structure of gut microbiota in CTX-induced immunosuppression. HWH regulated the gut microbiota by restoring *Lactobacillus*, *Lachnospiraceae*, *Christensenellaceae*, and *Bifidobacterium* and by decreasing *Ruminococcus*, *Staphylococcus*, and *Streptococcus* in CTX-treated mice. Additionally, HWH could increase the spleen indices and stimulate the secretions of TNF-α, IL-1β, and IL-6 in the colon. HWH improved the intestinal barrier, restored goblet cells in the crypt of the colon, and upregulated the expression of TJs. These findings suggest that HWH has potential as an effective immunomodulator, a prebiotic, and an intestinal mucosal repair agent. Detailed mechanisms of HWH on mitigating intestinal mucosal immunity and gut microbiota remain to be investigated. The development of sea cucumber-based functional foods is the future perspective in this field, and several factors, such as the organoleptic property, bio-accessibility, bioavailability, and personally designed products, need to be considered. Meanwhile, the production, extraction, and purification of active ingredients from sea cucumber at a large scale are other concerns for researchers.

## Figures and Tables

**Figure 1 ijms-24-12583-f001:**
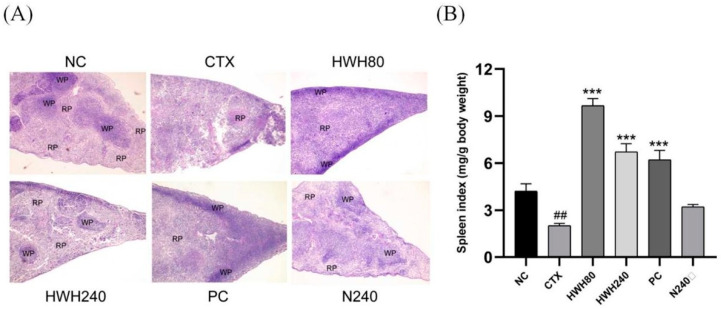
Effects of HWH on spleen histopathology and spleen indices in CTX-induced immunocompromised mice. Histopathology (magnification 40×): (**A**) spleen indices; (**B**) WP, white pulp and RP, red pulp; NC: normal control mice; CTX: mice treated with 80 mg/kg of CTX alone; HWH80: mice treated with CTX plus HWH (80 mg/kg); HWH240: mice treated with CTX plus HWH (240 mg/kg); PC: mice treated with CTX plus levamisole hydrochloride (40 mg/kg); N240: mice treated with HWH (240 mg/kg). Values are shown as mean ± SD (*n* = 10). ^##^ *p* < 0.01 compared to NC; *** *p* < 0.001 compared to CTX.

**Figure 2 ijms-24-12583-f002:**
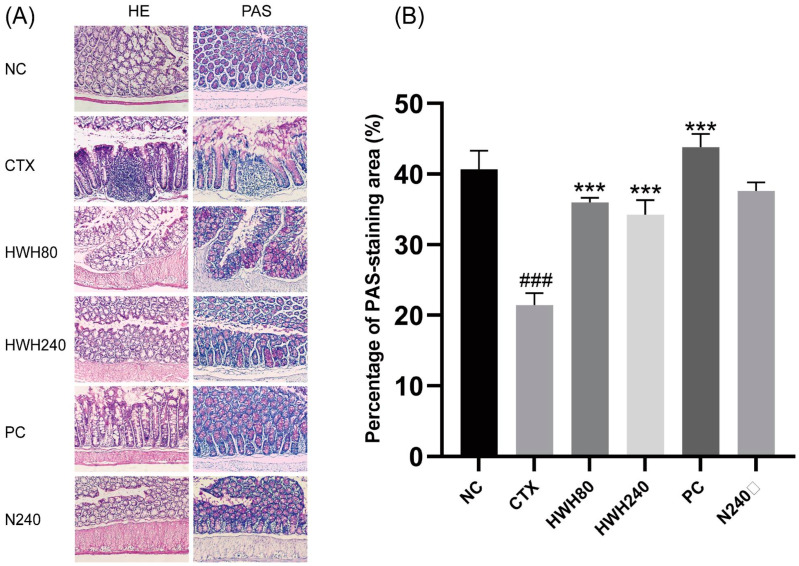
Effects of HWH on colon histopathology in CTX-induced immunocompromised mice. Representation of colon stained with H&E and PAS (magnification 100×) (**A**); percentage of PAS-staining area (%) (**B**). NC: normal control mice; CTX: mice treated with 80 mg/kg of CTX alone; HWH80: mice treated with CTX plus HWH (80 mg/kg); HWH240: mice treated with CTX plus HWH (240 mg/kg); PC: mice treated with CTX plus levamisole hydrochloride (40 mg/kg); N240: mice treated with HWH (240 mg/kg). Values are shown as mean ± SD (*n* = 10). *** *p* < 0.001 compared to the CTX group, and ^###^ *p* < 0.001 compared to the NC group.

**Figure 3 ijms-24-12583-f003:**
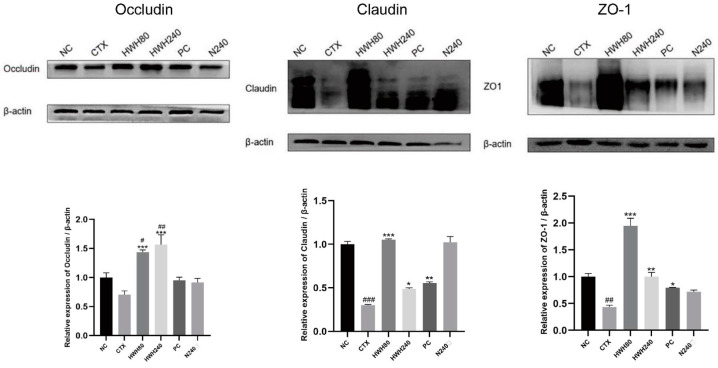
Effects of HWH on protein expressions of occludin, claudin, and ZO-1 in colon. NC: normal control mice; CTX: mice treated with 80 mg/kg of CTX alone; HWH80: mice treated with CTX plus HWH (80 mg/kg); HWH240: mice treated with CTX plus HWH (240 mg/kg); PC: mice treated with CTX plus levamisole hydrochloride (40 mg/kg); N240: mice treated with HWH (240 mg/kg). Values are shown as mean ± SD (*n* = 10). ^#^ *p* < 0.05, ^##^ *p* < 0.01, and ^###^ *p* < 0.001 compared with the NC group; * *p* < 0.05, ** *p* < 0.01, and *** *p* < 0.001 compared with the CTX group.

**Figure 4 ijms-24-12583-f004:**
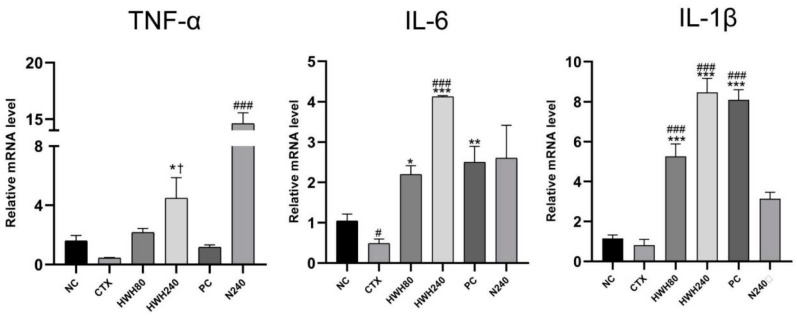
Effects of HWH on the mRNA levels of pro-inflammatory cytokines (IL-6, IL-1β, and TNF-α) in the colon. NC: normal control mice; CTX: mice treated with 80 mg/kg of CTX alone; HWH80: mice treated with CTX plus HWH (80 mg/kg); HWH240: mice treated with CTX plus HWH (240 mg/kg); PC: mice treated with CTX plus levamisole hydrochloride (40 mg/kg); N240: mice treated with HWH (240 mg/kg). Values are shown as mean ± SD (*n* = 10). ^#^ *p* < 0.05 and ^###^ *p* < 0.001 compared to NC; * *p* < 0.05, ** *p* < 0.01, and *** *p* < 0.001 compared to the CTX group; ^†^ *p* < 0.05 compared to the N240 group.

**Figure 5 ijms-24-12583-f005:**
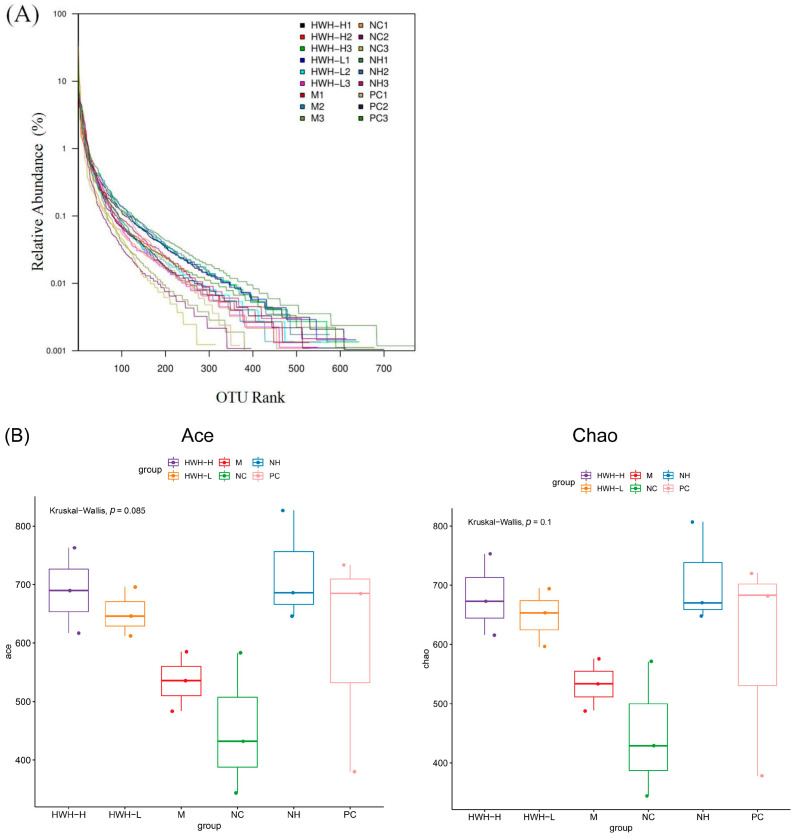
Effects of HWH on alpha diversity in CTX-induced immunocompromised mice. Rank abundance curve (**A**). Box plot of the gut microbiota (**B**). NC: normal control mice; M: mice treated with 80 mg/kg of CTX alone (CTX); HWH-L: mice treated with CTX plus HWH (80 mg/kg) (HWH80); HWH-H: mice treated with CTX plus HWH (240 mg/kg) (HWH240); PC: mice treated with CTX plus levamisole hydrochloride (40 mg/kg); NH: mice treated with HWH (240 mg/kg) (N240).

**Figure 6 ijms-24-12583-f006:**
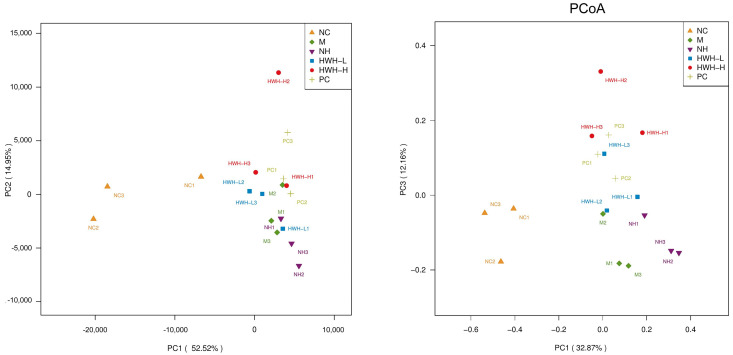
The effects of HWH on beta diversity in CTX-induced immunocompromised mice. NC: normal control mice; M: mice treated with 80 mg/kg of CTX alone (CTX); HWH-L: mice treated with CTX plus HWH (80 mg/kg) (HWH80); HWH-H: mice treated with CTX plus HWH (240 mg/kg) (HWH240); PC: mice treated with CTX plus levamisole hydrochloride (40 mg/kg); NH: mice treated with HWH (240 mg/kg) (N240).

**Figure 7 ijms-24-12583-f007:**
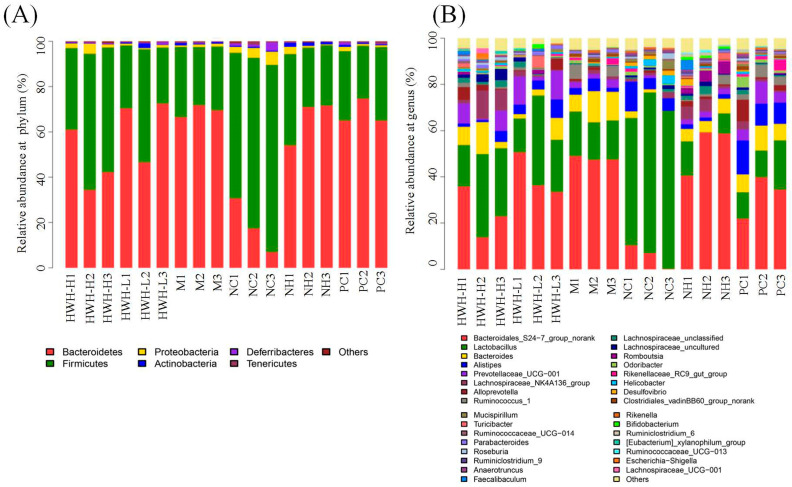
The effects of HWH on microbial community composition at different taxonomy levels in CTX-induced immunocompromised mice, with relative abundance (%) at the Phylum level (**A**) and Genus level (**B**). NC: normal control mice; M: mice treated with 80 mg/kg of CTX alone (CTX); HWH-L: mice treated with CTX plus HWH (80 mg/kg) (HWH80); HWH-H: mice treated with CTX plus HWH (240 mg/kg) (HWH240); PC: mice treated with CTX plus levamisole hydrochloride (40 mg/kg); NH: mice treated with HWH (240 mg/kg) (N240).

**Figure 8 ijms-24-12583-f008:**
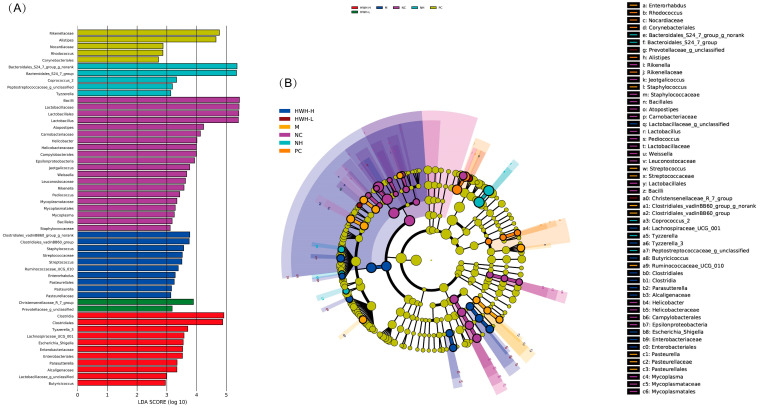
The effects of HWH on key phylotypes of gut microbiota in CTX-induced immunocompromised mice. LDA score (**A**); LEfSe taxonomic cladogram (**B**). NC: normal control mice; M: mice treated with 80 mg/kg of CTX alone (CTX); HWH-L: mice treated with CTX plus HWH (80 mg/kg) (HWH80); HWH-H: mice treated with CTX plus HWH (240 mg/kg) (HWH240); PC: mice treated with CTX plus levamisole hydrochloride (40 mg/kg); NH: mice treated with HWH (240 mg/kg) (N240).

**Table 1 ijms-24-12583-t001:** Determination of the HWH peptide sequence by LC-MS/MS.

Number	Sequence	Mass	Charges	Intensity	Activity Prediction Score
1	SRGLLSCLF	1051.55	2	371,770	0.83
2	GFDGPEGPRGPPGSE	1454.64	2	4,834,900	0.62
3	RGPAGPTGPTGPA	1134.57	2	2,150,400	0.56
4	AAVAAAVAPPSPPPIAGPP	1649.91	2	179,750	0.55
5	FDGPEGPRGPPGSE	1397.62	2	4,477,600	0.55
6	DLSEEFMAICSTMPDT	1845.75	2	2,482,400	0.42
7	SPGEKGDQGSPGPA	1282.58	2	1,278,600	0.38
8	VAPEEHPVLLTEAPLNPK	1953.06	2	158,170	0.34
9	FSGSQPELPVDQ	1302.61	2	8,266,800	0.26
10	KGADGETGEPGPQG	1298.57	2	273,750	0.22
11	LLSEMRRLE	1145.62	2	192,180	0.22
12	SYELPDGQVITIGNER	1789.88	2	765,850	0.21
13	PVSASRHQESANQGL	1579.77	2	6,068,000	0.20
14	ALAALQQSSSSGSSSSTAT	1739.82	2	454,080	0.19
15	GIVLDSGDGVTH	1168.57	2	1,385,600	0.18
16	LDLAGRDLTDY	1250.61	2	650,410	0.16
17	GEAGAETPKAATEAGEAP	1655.76	2	4,324,000	0.14
18	ATKYSDITKLSSIGKSVE	1926.03	2	5,239,900	0.09
19	YAYSVKNAVQDAP	1424.69	2	302,260	0.09
20	DVITIEVLAK	1099.64	2	2,187,900	0.08
21	MEHDTRTHREHYR	1766.80	2	13,898,000	0.06
22	EKLCYVALDFEQEMATAASSSSLEK	2806.30	3	335,090	0.04
23	LCYVALDFEQEMATAASSSSLEK	2549.17	3	23,738,000	0.03

G-Glycine, P-Proline, Q-Glutamine, W-Tryptophan, L-Leucine, D-Aspartic acid, F-Phenylalanine, T-Threonine, M-Methionine, A-Alaninie, R-Arginine, S-Serine, I-Isoleucine, K-Lysine.

## Data Availability

The original contributions presented in the study are included in the article/Appendix A; further inquiries can be directed to the corresponding authors.

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
