# Peer review of "Holothurian Wall Hydrolysate Ameliorates Cyclophosphamide-Induced Immunocompromised Mice via Regulating Immune Response and Improving Gut Microbiota"

_ijms, 2023, doi:10.3390/ijms241612583_

Round 1

Reviewer 1 Report

It is a very well done article from the point of view of research methodology. In my opinion, the article could be improved by taking into account the presence of boron in the hydrolyzate, the latest research shows that boron is an essential element in healthy symbiosis. The dysbiosis improvement effects can also be due to the high boron content in the hydrolyzate, seeing that Sea cucumber has a high boron content over 50 ppm!

Reviewer 2 Report

This manuscript presented an investigation of the role of holothurian wall hydrolysates in injury repair to intestinal barrier and maintanance of gut microbiota balance. The work shows the value of holothurian wall hydrolysates. However, it was presented with a series of flaws:

1. Figure 1 doesn't show any meaningful data.

2. Table 1 needs explanation on how it was generated. And in what order the data is organized.

3. In line 142-144, it is obvious from Figure 2B that NC (also N240) shows 2-fold increase in spleen index compared to CTX (and N240 may be 1.5-fold). What is the criterion here to define significance?

4. In Section 2.3 and Figure 3, Figure 3A needs to be re-organized into 2-columns by 6-rows (HE/PAS as column title). The HE staining clearly shows PC is very similar to CTX as opposed to the statement in line 161.

5. In Section 2.4 and Figure 4, Figure 4A(panel 2 and 3) shows NC is significantly different than N240 again in contrary to the authors' statement (line 183). And western blots (panel 2 and 3) don't match H&E in Figure 3A. Figure 4 is missing figure title and subtitles.

Minor:

6. Figure 6,7,10 need higher resolution.

7. Line 11, "...closely related to the integrity of intestinal barrier and the balance of gut microbiota".

8. "Ameliorates".

Needs more proofreading

Reviewer 3 Report

The manuscript describes the amelioration of cyclophosphamide-induced immunocompromised mice via regulating immune response and improving gut microbiota by holothurian wall hydrolysates. The topic is relevant to the aim and scope of the Int. J. Mol. Sci.. The manuscript is well written and easy to follow. Some clarifications in the texts are needed. Overall, this manuscript meets the standard for acceptance after addressing the below comments:

1)      Since there are a lot of abbreviations used in this manuscript, it would be better to include a separate section for the abbreviation, for example, ZO-1, PC, TRL, MAPK, PI3K, AKT, and so on. Actually these are quite familiar to the biologists, but research papers should be general to all readers.

2)      Figures are little recognizable, especially Figure 6, 7, and, 9. Please modify them.

3)      What is the variable on the vertical axis in Figure 6? The variable looks missing.

4)      In Figure 4, the effect of HWH is presented with the amount of the protein expression. Then, why is the effect presented in Figure 5 with the relative mRNA level rather than the expression?

5)      Why is the relative mRNA level of TNF-α in Figure 5 extraordinary high at N240, although IL-6 and IL-1β is not quite high?

6)      Please elaborate the experimental steps for staining. What is the dye concentration and its volume used to the slide? What is the staining time after the dye solution was added?

Round 2

Reviewer 3 Report

The response 6 should be included in the manuscript. The other issues have been addressed.
